# Universal Sequence Preconditioning

Annie Marsden [*]                Elad Hazan[*][†]

## Abstract

We study the problem of preconditioning in sequential prediction. From the theoretical lens of linear dynamical systems, we show that convolving the target sequence corresponds to applying a polynomial to the hidden transition matrix. Building on this insight, we propose a universal preconditioning method that convolves the target with coefficients from orthogonal polynomials such as Chebyshev or Legendre. We prove that this approach reduces regret for two distinct prediction algorithms and yields the first ever sublinear and hidden-dimension–independent regret bounds (up to logarithmic factors) that hold for systems with marginally stable and asymmetric transition matrices. Finally, extensive synthetic and real-world experiments show that this simple preconditioning strategy improves the performance of a diverse range of algorithms, including recurrent neural networks, and generalizes to signals beyond linear dynamical systems.

## 1 Introduction

In sequence prediction the goal of the learner is to predict the next token accurately according to a specified loss function, such as the mean square error or cross-entropy. This fundamental problem in machine learning has gained increased importance with the rise of large language models, which perform sequence prediction on tokens using cross entropy. The focus of this paper is *preconditioning*, i.e. modifying the target sequence to make it easier to learn. A classic example is **differencing**, introduced by Box and Jenkins in the 1970s [13], which transforms observations $\mathbf{y}_1, \mathbf{y}_2, \ldots$ into successive differences,

$$\mathbf{y}_1 - \mathbf{y}_0, \ \mathbf{y}_2 - \mathbf{y}_1, \ ..., \ \mathbf{y}_t - \mathbf{y}_{t-1}, ...$$

It is widely acknowledged that learning this sequence can be "easier" than learning the original sequence for a large number of modalities. In this work we seek a more general framework for sequence preconditioning that captures the same intuition behind differencing and extends it to a broader class of transformations. The question we ask is

*What is the general form of sequence preconditioning that enables provably accurate learning?*

We address this question by introducing a preconditioning method which takes in $n$ fixed coefficients $c_0, \ldots, c_n$ and converts the sequence of observations $\mathbf{y}_1, \ldots, \mathbf{y}_t, \ldots$ to the sequence of convolved observations[1]

$$c_0\mathbf{y}_0, \ c_0\mathbf{y}_1 + c_1\mathbf{y}_0, \ \ldots, \ \sum_{i=0}^{n} c_i\mathbf{y}_{t-i}, \ \ldots$$

From an information-theoretic perspective, approaches of this kind seem futile—predicting $\mathbf{y}_t$ or $\sum_i c_i\mathbf{y}_{t-i}$ seems equally hard in an adversarial setting. Yet we show that when the data arises from a linear dynamical system (LDS), there exists a *universal* form of preconditioning that provably

---

[*]Google Deepmind
[†]Princeton University

[1]This recovers differencing when $n = 2$, $c_0 = 1$, and $c_1 = -1$.

39th Conference on Neural Information Processing Systems (NeurIPS 2025).

improves learnability, independent of the specific system. In the LDS setting, we show that preconditioning significantly strengthens existing prediction methods, leading to new regret bounds. Here, preconditioning has an elegant effect: the preconditioning filter forms coefficients of an $n$ degree polynomial, and the hidden system transition matrix is evaluated on this polynomial– potentially shrinking the domain. In this setting, shrinking the learnable domain is akin to making the problem "easier to learn", a relationship that is formalized by [26]. This allows us to prove the first dimension-independent sublinear regret bounds for asymmetric linear dynamical systems that are marginally stable.

## 1.1 Our results

Our main contribution is *Universal Sequence Preconditioning*, a novel method of sequence preconditioning which convolves the target sequence with the coefficients of the $n$-th monic Chebyshev polynomial. We give a more general form of preconditioning, allowing arbitrary user-specified coefficients, in Algorithm 1 and an online version in Algorithm 4 (Appendix C). We analyze the effect of Universal Sequence Preconditioning on two canonical sequence prediction algorithms in the online setting: (1) convex regression and (2) spectral filtering. In either case, the results are impressive– yielding the first known sublinear regret bounds as compared to the optimal ground-truth predictor that are simultaneously (1) applicable to marginally stable systems, (2) independent of the hidden dimension (up to logarithmic factors), and (3) applicable to systems whose transition matrix is asymmetric[2] (see Table 1 ).

---

**Algorithm 1** General Sequence Preconditioning (Offline Version)

---

1: **Training**
2: Input: training data $(\mathbf{u}_{1:T}^{1:N}, \mathbf{y}_{1:T}^{1:N})$ where $(\mathbf{u}_t^i, \mathbf{y}_t^i)$ is the $t$-th input/output pair in the $i$-th sequence; coefficients $c_{0:n}$; prediction algorithm $\mathcal{A}$.
3: Assert $c_0 = 1$.
4: **for** $i = 1$ to $N$ **do**
5: $\quad \mathbf{y}_{1:T}^{\text{preconditioned},i} \leftarrow \text{convolution}(\mathbf{y}_{1:T}^i, c_{0:n})$ $\qquad \triangleleft \mathbf{y}_t^{\text{preconditioned},i} = \mathbf{y}_t^i + \sum_{j=1}^n c_j \mathbf{y}_{t-j}^i$
6: **end for**
7: Train $\mathcal{A}$ on preconditioned data $\left(\mathbf{u}_{1:T}^{1:N}, \mathbf{y}_{1:T}^{\text{preconditioned},1:N}\right)$.
8: **Test Time**
9: **for** $t = 1$ to $T$ **do**
10: $\quad$ Receive $\mathbf{u}_t$.
11: $\quad$ Predict $\hat{\mathbf{y}}_t \leftarrow \mathcal{A}\left(\mathbf{u}_{1:t}, \mathbf{y}_{1:(t-1)}\right) - \sum_{i=1}^n c_i \mathbf{y}_{t-i}$.
12: $\quad$ Receive $\mathbf{y}_t$.
13: **end for**

---

First, applying USP to standard convex regression results in regret $\tilde{O}(T^{-2/13})$, which holds simultaneously across the three settings above and remains dimension-independent. For comparison, a naive analysis of regression yields a vacuous regret bound of $O(T^{5/2})$ on marginally stable systems. Second, combining USP with a variant of *spectral filtering* [27] that uses novel filters, the algorithm is able learn a broader class of linear dynamical systems– in particular systems whose hidden transition matrix may be asymmetric. The enhanced method achieves regret $\tilde{O}(T^{-3/13})$, the best known rate under the joint conditions of (1)-(3) discussed above: marginal stability, dimension independence, and asymmetry. Both results require that the transition matrix eigenvalues have imaginary parts bounded by $O(1/\log T)$—a near-tight condition for achieving dimension-free regret. Further discussion on this appears in Appendix B.

Empirical results in Section 4 demonstrate that USP consistently improves performance across diverse algorithms—including regression, spectral filtering, and neural networks—and across data types extending beyond linear dynamical systems.

---

[2]Our results only hold for asymmetric matrices whose eigenvalues have imaginary component bounded above by $O(1/\log(T))$. This is somewhat tight, see Section B.

| Method | Marginally stable | $d_{\text{hidden}}$-free | Asymmetric |
|---|---|---|---|
| Sys-ID | × | × | ✓ |
| Regression (open-loop) | × | ✓ | ✓ |
| Regression (closed-loop) | ✓ | × | ✓ |
| Spectral Filtering | ✓ | ✓ | × |
| USP + Regression | ✓ | ✓ | ∼ ✓ |
| USP + Spectral Filtering | ✓ | ✓ | ∼ ✓ |

Table 1: Comparison of methods for learning LDS. USP extends learning to mildly asymmetric matrices with bounded complex eigenvalues.

## 1.2 Intuition for Universal Sequence Preconditioning

We now give some brief intuition for the result. Linear dynamical systems (LDS) are perhaps the most basic and well studied dynamical systems in engineering and control science. Given input vectors $\mathbf{u}_1, \ldots, \mathbf{u}_T \in \mathbb{C}^{d_{\text{in}}}$, the system generates a sequence of output vectors $\mathbf{y}_1, \ldots \mathbf{y}_T \in \mathbb{C}^{d_{\text{out}}}$ according to the law

$$\mathbf{x}_{t+1} = \mathbf{A}\mathbf{x}_t + \mathbf{B}\mathbf{u}_t, \qquad \mathbf{y}_t = \mathbf{C}\mathbf{x}_t + \mathbf{D}\mathbf{u}_t, \tag{1}$$

where $\mathbf{x}_0, \ldots, \mathbf{x}_T \in \mathbb{C}^{d_{\text{hidden}}}$ is a sequence of hidden states and $(\mathbf{A}, \mathbf{B}, \mathbf{C}, \mathbf{D})$ are matrices which parameterize the LDS. We assume w.l.o.g. that $\mathbf{D} = 0$. We can factor out the hidden state $\mathbf{x}_t$ so that the observation at time $t$ is

$$\mathbf{y}_t = \sum_{s=1}^{t} \mathbf{C}\mathbf{A}^{t-s}\mathbf{B}\mathbf{u}_s.$$

Given coefficients $c_{0:n} = (c_0, \ldots, c_n)$ let

$$p_n^c(x) \stackrel{\text{def}}{=} \sum_{i=0}^{n} c_i x^{n-i}. \tag{2}$$

Consider a "preconditioned" target at time $t$ to be a linear combination of $\mathbf{y}_{t:t-n}$ with coefficients $c_{0:n}$. A key insight is the following identity,

$$\sum_{i=0}^{n} c_i \mathbf{y}_{t-i} = \sum_{s=0}^{n-1} \left( \sum_{i=0}^{s} c_i \mathbf{C}\mathbf{A}^{s-i}\mathbf{B} \right) \mathbf{u}_{t-s} + \sum_{s=0}^{t-n-1} \mathbf{C}p_n^c(\mathbf{A})\mathbf{A}^s\mathbf{B}\mathbf{u}_{t-n-s}.$$

If we take $c_0 = 1$ (i.e. a monic polynomial), we can re-write $\mathbf{y}_t$ as

$$\mathbf{y}_t = \underbrace{-\sum_{i=1}^{n} c_i \mathbf{y}_{t-i}}_{\aleph_0} + \underbrace{\sum_{s=0}^{n-1} \sum_{i=0}^{s} c_i \mathbf{C}\mathbf{A}^{s-i}\mathbf{B}\mathbf{u}_{t-s}}_{\aleph_1} + \underbrace{\sum_{s=0}^{t-n-1} \mathbf{C}p_n^c(\mathbf{A})\mathbf{A}^s\mathbf{B}\mathbf{u}_{t-n-s}}_{\aleph_2}. \tag{3}$$

This expression highlights our approach as a balance of three terms:

$\aleph_0$ The **universal preconditioning** term: it depends only on the coefficients $c_{0:n}$ and not on any learning algorithm.

$\aleph_1$ A term learnable via convex relaxation and regression, for example by denoting

$$\mathbf{Q}_s^{\text{learned}} = \sum_{i=0}^{s} c_i \mathbf{C}\mathbf{A}^{s-i}\mathbf{B}.$$

The diameter of the coefficient $\mathbf{Q}_s$ depends on the magnitude of the coefficients $c_{0:n}$.

$\aleph_2$ The residual term with polynomial $p_n^c(\mathbf{A})$. By a careful choice of coefficients $c_{0:n}$, we can force this term to be very small.

The main insight we derive from this expression is the inherent tension between two terms $\aleph_1, \aleph_2$. The polynomial $p_n^c(x)$ and its coefficients $c_0, \ldots, c_n$ control two competing effects:

1. The preconditioning coefficients grow larger with the degree $n$ of the polynomial and the magnitude of the coefficients $c_i$. A higher degree polynomial and larger coefficients increase the diameter of the search space over the preconditioning coefficients, and therefore increase the regret bound stemming from the $\aleph_1$ component learning.

2. On the other hand, a larger search space can allow a broader class of polynomials $p_n(\cdot)$ which can better control of the magnitude of $p_n(\mathbf{A})$, and therefore reduce the search space of the $\aleph_2$ component.

What choice of polynomial is best? This work considers the Chebyshev polynomial. The reason is the following property of the $n$-th monic Chebyshev polynomial:

$$\max_{\lambda \in [-1,1]} |p_n(\lambda)| \leq 2^{-(n-1)}.$$

As an example, consider any LDS whose hidden transition matrix $\mathbf{A}$ is diagonalizable and has eigenvalues in $[-1, 1]$[3]. By the above property, observe that $\|p_n(\mathbf{A})\|_\infty \leq 2 \cdot 2^{-n}$, therefore shrinking the $\aleph_2$ term at a rate exponential with the number of preconditioning coefficients. We pause to remark on the *universality* of this choice of polynomial. Indeed, one could instead have chosen the preconditioning coefficients to depend on $\mathbf{A}$ so that $p_n^{\mathbf{c}}(\cdot)$ is the characteristic polynomial of $\mathbf{A}$. By the Cayley-Hamilton theorem, $p_n(\mathbf{A}) = 0$. This means that $\aleph_2$ term is canceled out completely. However this would have required knowledge of the spectrum of $\mathbf{A}$. The Chebyshev polynomial, on the other hand, is agnostic to the particular hidden transition matrix. Moreover, even if the spectrum of $\mathbf{A}$ were known, choosing the preconditioning coefficients to form the characteristic polynomial would result in an algorithm which must learn hidden dimension many parameters, which is prohibitive. Instead, the degree of the Chebyshev polynomial must only grow logarithmically with the hidden dimension.

## 1.3 Related work

Our manuscript is technically involved and incorporates linear dynamical systems, spectral filtering, complex Chebyshev and Legendre polynomials, Hankel and Toeplitz matrix eigendecay, Gaussian quadrature and other techniques. The related work is thus expansive, and due to space limitations we give a detailed treatment in Appendix A. Preconditioning in the context of time series analysis has roots in the classical work of Box and Jenkins [13]. In their foundational text they propose differencing as a method for making the time series stationary, and thus amenable to statistical learning techniques such as ARMA (auto-regressive moving average) [9]. The differencing operator can be applied numerous times, and for different lags, giving rise to the ARIMA family of forecasting models. Identifying the order of an ARIMA model, and in particular the types of differencing needed to make a series stationary, is a hard problem. This is a special case of the problem we consider: differencing corresponds to certain coefficients of preconditioning the time series, whereas we consider arbitrary coefficients. For a thorough introduction to modern control theory and exposition on open loop / closed loop predictors, learning via regression, and spectral filtering, see [26]. The fundamental problem of learning in linear dynamical systems has been studied for many decades, and we highlight several key approaches below:

1. System identification refers to the method of recovering $\mathbf{A}, \mathbf{B}, \mathbf{C}$ from the data. This is a non-convex problem and while many methods have been considered in this setting, they depend polynomially on the hidden dimension.

2. The (auto) regression method predicts according to $\hat{\mathbf{y}}_t = \sum_{i=1}^{h} \mathbf{M}_i \mathbf{u}_{t-i}$. The coefficients $\mathbf{M}_i$ can be learned using convex regression. The downside of this approach is that if the spectral radius of $\mathbf{A}$ is $1 - \delta$, it can be seen that $\sim \frac{1}{\delta}$ terms are needed.

3. The regression method can be further enhanced with "closed loop" components, that regress on prior observations $\mathbf{y}_{t-1:1}$. It can be shown using the Cayley-Hamilton theorem that using this method, $d_h$ components are needed to learn the system, where $d_h$ is the *hidden* dimension of $\mathbf{A}$.

4. Filtering involves recovering the state $x_t$ from observations. While Kalman filtering is optimal under specific noise conditions, it generally fails in the presence of marginal stability and adversarial noise.

---

[3]This work considers a broader class of hidden transition matrices.

5. Finally, spectral filtering combines the advantages of all methods above. It is an efficient method, its complexity does **not** depend on the hidden dimension, and works for marginally stable systems. However, spectral filtering requires $\mathbf{A}$ to be symmetric, or diagonalizable under the real numbers.

## 2 Main Results

In this section we formally state our main algorithms and theorems. We show that the Universal Sequence Preconditioning method provides significantly improved regret bounds for learning linear dynamical systems than previously known when used in conjunction with two distinct methods. The first method is simple convex regression, and the second is spectral filtering. Both algorithms allow for learning in the case of marginally stable linear dynamical systems and allow for certain asymmetric transition matrices of arbitrary high hidden dimension. The regret bounds are free of the hidden dimension (up to logarithmic factors) – which significantly extends the state of the art.

### 2.1 Universal Sequence Preconditioning Applied to Regression

Algorithm 2 is an instantiation of Algorithm 4 for the method of convex regression. We set the preconditioning coefficients to be the coefficients of the $n$-th degree (monic) Chebyshev polynomial.

---

**Algorithm 2** Universal Sequence Preconditioning for Regression

---

1: Input: initial parameter $\mathbf{Q}^0$; preconditioning coefficients $\mathbf{c}_{0:n}$ from the $n$-th degree (monic) Chebyshev polynomial; convex constraints $\mathcal{K} = \{(\mathbf{Q}_0, \ldots, \mathbf{Q}_{n-1})$ s.t. $\|\mathbf{Q}_j\| \leq C_{\text{domain}}\|\mathbf{c}\|_1\}$
2: Assert that $\mathbf{c}_0 = 1$.
3: **for** $t = 1$ to $T$ **do**
4:     Receive $\mathbf{u}_t$.
5:     Predict $\hat{\mathbf{y}}_t(\mathbf{Q}^t) = -\sum_{i=1}^n \mathbf{c}_i \mathbf{y}_{t-i} + \sum_{j=0}^n \mathbf{Q}_j^t \mathbf{u}_{t-j}$.
6:     Observe true output $\mathbf{y}_t$ and suffer loss $\ell_t(\mathbf{Q}^t) = \|\hat{\mathbf{y}}_t(\mathbf{Q}^t) - \mathbf{y}_t\|_1$.
7:     Update and project:
$$\mathbf{Q}^{t+1} \leftarrow \text{proj}_{\mathcal{K}} \left( \mathbf{Q}^t - \eta_t \nabla_{\mathbf{Q}} \ell_t(\mathbf{Q}^t) \right).$$

8: **end for**

---

Theorem 2.1 shows that vanishing loss compared to the optimal ground-truth predictor, at a rate that is independent of the hidden dimension of the system.

**Theorem 2.1.** *Let $\{\mathbf{u}_t\}_{t=1}^T \in \mathbb{C}^{d_{in}}$ be any sequence of inputs which satisfy $\|\mathbf{u}_t\|_2 \leq 1$ and let $\{\mathbf{y}_t\}_{t=1}^T \in \mathbb{C}^{d_{out}}$ be the corresponding output coming from some linear dynamical system $(\mathbf{A}, \mathbf{B}, \mathbf{C})$ as defined per Eq. 1. Let $\mathbf{P}$ diagonalize $\mathbf{A}$ (note $\mathbf{P}$ exists w.l.o.g.) and let $\kappa = \|\mathbf{P}\|\|\mathbf{P}^{-1}\|$. Assume that $\|\mathbf{B}\|\|\mathbf{C}\|\kappa \leq C_{domain}$. Let $\lambda_1, \ldots, \lambda_{d_h}$ denote the spectrum of $\mathbf{A}$. If*

$$\max_{j \in [d_h]} |\arg(\lambda_j)| \leq 1/(32 \log_2(2T^3/d_{out}))^2$$

*then the predictions $\hat{\mathbf{y}}_1, \ldots, \hat{\mathbf{y}}_T$ from Algorithm 2 where the preconditioning coefficients $\mathbf{c}_{0:n}$ are chosen to be the coefficients of the $n$-th monic Chebyshev polynomial satisfy*

$$\frac{1}{T} \sum_{t=1}^T \|\hat{\mathbf{y}}_t - \mathbf{y}_t\|_1 \leq \tilde{O}\left( \frac{\|\mathbf{B}\|\|\mathbf{C}\|\kappa\sqrt{d_{out}}}{T^{2/13}} \right),$$

*where $\tilde{O}(\cdot)$ hides polylogarithmic factors in $T$.*

The proof of Theorem 2.1 is in Appendix D. For a simple baseline comparison, the regret achieved (via the same proof technique) by the vanilla regression algorithm without preconditioning is $O\left(C_{\text{domain}}\sqrt{d_{\text{out}}}T^{5/2}\right)$ which is not sublinear in $T$.

### 2.2 Universal Sequence Preconditioning Applied to Spectral Filtering

Our second main result is the application of Universal Sequence Preconditioning to the spectral filtering algorithm [27]. Our results are more general and apply to any choice of polynomial, not

just Chebyshev. In addition to applying USP to spectral filtering, we also propose a novel spectral filtering basis. Both changes to the vanilla spectral filtering algorithm are necessary to extend its sublinear regret bounds to the case of underlying systems with asymmetric hidden transition matrices. First we define the spectral domain

$$\mathbb{C}_\beta = \{z \in \mathbb{C} \mid |z| \leq 1, |\arg(z)| \leq \beta\}.$$

Given horizon $T$ and $\alpha \in \mathbb{C}_\beta$ let

$$\tilde{\mu}_T(\alpha) \overset{\text{def}}{=} (1 - \alpha^2) \begin{bmatrix} 1 & \alpha & \dots & \alpha^{T-1} \end{bmatrix}^\top, \tag{4}$$

and

$$\mathbf{Z}_T \overset{\text{def}}{=} \int_{\alpha \in \mathbb{C}_\beta} \tilde{\mu}_T(\alpha) \tilde{\mu}_T(\overline{\alpha})^\top d\alpha, \tag{5}$$

where $\overline{\alpha} \in \mathbb{C}$ denotes the complex conjugate. The novel spectral filters are the eigenvectors of $\mathbf{Z}_{T-n-1}$, which we denote as $\phi_1, \dots, \phi_{T-n-1}$. Note that in the standard spectral filtering literature, the spectral filtering matrix is an integral over the real line and does not involve the complex conjugate. Our new matrix has an entirely different structure and although it looks quite similar, it surprisingly upends the proof techniques to ensure exponential spectral decay, a critical property for the method. Future work examines this matrix more thoroughly, but in this paper we simply provide a standard bound on its eigenvalues.

---

**Algorithm 3** Universal Sequence Preconditioning for Spectral Filtering

---

1: Input: initial $\mathbf{Q}_{1:n}^1, \mathbf{M}_{1:k}^1$, horizon $T$, convex constraints

$$\mathcal{K} = \{(\mathbf{Q}_0, \dots, \mathbf{Q}_{n-1}, \mathbf{M}_1, \dots, \mathbf{M}_k \text{ s.t. } \|\mathbf{Q}_j\| \leq R_Q \text{ and } \|\mathbf{M}_j\| \leq R_M\},$$

   parameter $n$, coefficients $\mathbf{c}_{1:n}$.
2: Let $p_n^{\mathbf{c}}(x) = \mathbf{c}_0 x^n + \mathbf{c}_1 x^{n-1} + \cdots + \mathbf{c}_n$ and $\tilde{p}_n^{\mathbf{c}}(x) = (1 - x^2)p_n^{\mathbf{c}}(x)$. Let $\tilde{\mathbf{c}}_0, \dots, \tilde{\mathbf{c}}_{n+2}$ be the coefficients of $\tilde{p}_n^{\mathbf{c}}(x)$.
3: Let $\phi_1, \dots, \phi_n$ be the top $n$ eigenvectors of $\mathbf{Z}_{T-n-1}$.
4: Assert $\tilde{\mathbf{c}}_0 = 1$.
5: **for** $t = 1$ to $T$ **do**
6:    Let $\tilde{\mathbf{u}}_{t-n-1:1}$ be $\mathbf{u}_{t-n-1:1}$ padded with zeros so it has dimension $T - n - 1 \times d_{\text{in}}$.
7:    Predict $\hat{\mathbf{y}}_t(\mathbf{Q}^t, \mathbf{M}^t) = -\sum_{i=1}^{n+2} \tilde{\mathbf{c}}_i \mathbf{y}_{t-i} + \sum_{j=0}^n \mathbf{Q}_j^t \mathbf{u}_{t-j} + \frac{1}{\sqrt{T}} \sum_{j=1}^k \mathbf{M}_j^t \phi_j^\top \tilde{\mathbf{u}}_{t-n-1:1}$.
8:    Observe true $\mathbf{y}_t$, define loss $\ell_t(\hat{\mathbf{y}}_t) = \|\hat{\mathbf{y}}_t(\mathbf{Q}^t, \mathbf{M}^t) - \mathbf{y}_t\|_1$.
9:    Update and project: $(\mathbf{Q}^{t+1}, \mathbf{M}^{t+1}) = \text{proj}_{\mathcal{K}}(\mathbf{Q}^t, \mathbf{M}^t) - \eta_t \nabla \ell_t(\mathbf{Q}^t, \mathbf{M}^t))$
10: **end for**

---

**Theorem 2.2.** *Let $\{\mathbf{u}_t\}_{t=1}^T \in \mathcal{R}^{d_{in}}$ be any sequence of inputs which satisfy $\|\mathbf{u}_t\|_2 \leq 1$ and let $\{\mathbf{y}_t\}_{t=1}^T$ be the corresponding output coming from some linear dynamical system $(\mathbf{A}, \mathbf{B}, \mathbf{C})$ as defined per Eq. 1. Let $\mathbf{P}$ diagonalize $\mathbf{A}$ (note $\mathbf{P}$ exists w.l.o.g.) and let $\kappa = \|\mathbf{P}\|\|\mathbf{P}^{-1}\|$. Suppose the radius parameters of Algorithm 3 satisfy $R_Q \geq \|\mathbf{C}\|\|\mathbf{B}\|\|\mathbf{c}\|_1$ and $R_M \geq 2\|\mathbf{C}\|\|\mathbf{B}\|\kappa \log(T) \left(\max_{j \in [d_h]} |\arg(\lambda_j)|\right)^{4/3} T^{7/6} \left(\max_{\alpha \in \mathbb{C}_\beta} |p_n^{\mathbf{c}}(\alpha)|\right)$. Further suppose that the eigenvalues of $\mathbf{A}$ have bounded argument:*

$$\max_{j \in [d_h]} |\arg(\lambda_j)| \leq T^{-1/3}.$$

*Then the predictions $\hat{\mathbf{y}}_1, \dots, \hat{\mathbf{y}}_T$ from Algorithm 3 where the preconditioning coefficients $\mathbf{c}_{0:n}$ are chosen to be the coefficients of the $n$-th monic Chebyshev polynomial satisfy*

$$\frac{1}{T} \sum_{t=1}^T \|\hat{\mathbf{y}}_t - \mathbf{y}_t\|_1 \leq \tilde{O}\left(\frac{\|\mathbf{C}\|\|\mathbf{B}\|\kappa\sqrt{d_{out}}}{T^{1/39}}\right).$$

We remark that the result of Theorem 2.2 is rather weak. Although the complex eigenvalues of $\mathbf{A}$ are not trivially bounded (trivial would be a bound of $1/T$), they still must be polynomially small in $T$. Moreover we note that the proof technique for the result does not make use of the critical properties of spectral filtering and relies much more on the power of preconditioning. The proof of Theorem 2.2 is in Section E.

# 3 Proof Overview

In this section we give a high level overview of the proofs for Theorem 2.1 and Theorem 2.2. We start by recalling the intuition for Universal Sequence Preconditioning developed in Section 1.2 which shows that if $\{\mathbf{y}_t\}_{t=1}^{T}$ evolves as a linear dynamical system parameterized by matrices $(\mathbf{A}, \mathbf{B}, \mathbf{C})$ with inputs $\{\mathbf{u}_t\}_{t=1}^{T}$ then by Equation 3,

$$\mathbf{y}_t = \underbrace{-\sum_{i=1}^{n} c_i \mathbf{y}_{t-i}}_{\aleph_0} + \underbrace{\sum_{s=0}^{n-1}\sum_{i=0}^{s} c_i \mathbf{C}\mathbf{A}^{s-i}\mathbf{B}\mathbf{u}_{t-s}}_{\aleph_1} + \underbrace{\sum_{s=0}^{t-n-1} \mathbf{C}p_n^c(\mathbf{A})\mathbf{A}^s\mathbf{B}\mathbf{u}_{t-n-s}}_{\aleph_2}.$$

Recall that $\aleph_0$ is the universal preconditioning component, $\aleph_1$ is the term that can easily be learned by convex relaxation and regression, and $\aleph_2$ is the critical term that contains $p_n^c(\mathbf{A})$. Both Theorem 2.1 and Theorem 2.2 use the standard result from online convex optimization (Theorem 3.1 from [24]) that online gradient descent over convex domain $\mathcal{K}$ achieves regret $\frac{3}{2}GD\sqrt{T}$ as compared to the best point in $\mathcal{K}$, where $D$ denotes the diameter of $\mathcal{K}$ and $G$ denotes the maximum gradient norm.

**Regression: Proof of Theorem 2.1**  In the case of regression, the domain is chosen so that $\aleph_2$ may be learned and the proof proceeds by bounding the diameter of such a domain and its corresponding maximum gradient norm to get regret $Cn^2\sqrt{d_{\text{out}}}\|c\|_1\sqrt{T}$ for a universal constant $C > 0$ which depends on the norms of matrices $\mathbf{B}$ and $\mathbf{C}$ from the underlying system. Then $\aleph_3$ is treated as an un-learnable error term. Let $\lambda(\mathbf{A})$ denote the set of eigenvalues of $\mathbf{A}$. By the simple magnitude bound of

$$\|\sum_{s=0}^{t-n-1} \mathbf{C}p_n(\mathbf{A})\mathbf{A}^s\mathbf{B}\mathbf{u}_{t-n-s}\| \leq \max_{\lambda \in \lambda(\mathbf{A})} |p_n(\lambda)| \cdot T \cdot \|\mathbf{C}\| \cdot \|\mathbf{B}\|,$$

the error of ignoring this term can be very small if $\max_{\lambda(\mathbf{A})} |p_n(\lambda(\mathbf{A}))|$ is small. In the proof of Theorem 2.1 in Appendix D we show that the regret for a generic polynomial $p_n^c$ defined by coefficients $c_{0:n}$ is

$$\sum_{t=1}^{T} \|\mathbf{y}_t - \hat{\mathbf{y}}_t\|_1 \leq \underbrace{Cn^2\sqrt{d_{\text{out}}}\|c\|_1\sqrt{T}}_{\text{Regret from learning } \aleph_2} + \underbrace{C\max_{\lambda \in \mathcal{D}} |p_n^c(\lambda)|T^2}_{\text{Unlearnable Error Term}},$$

where $\mathcal{D}$ is the region where $\mathbf{A}$ is allowed to have eigenvalues (see Theorem D.1). Therefore, to get sublinear regret, we must choose a polynomial which has bounded $\ell_1$ norm of its coefficients, while also exhibits very small infinity norm on the domain of $\mathbf{A}$'s eigenvalues.

**Spectral Filtering: Proof of Theorem 2.2**  In the case of spectral filtering, the domain is chosen so that both $\aleph_2$ and $\aleph_3$ may be learned. Because spectral filtering learns $\aleph_3$, it is able to accumulate less error and hence achieves a better regret bound of $O(T^{-3/13})$ as compared to regression's $O(T^{-2/13})$. At a high level, the proof proceeds by exploiting the fact that $p_n(\mathbf{A})$ shrinks the size of the learnable domain. However this is not enough, in order to extend the result to systems where $\mathbf{A}$ may have complex eigenvalues, the spectral filters must be eigenvalues of a new matrix, defined in Eq. 5, whose domain of integration includes the possibly complex eigenvalues of $\mathbf{A}$. To get the dimension-independent regret bounds enjoyed by spectral filtering in this new setting where complex eigenvalues may occur, the exponential decay of $\mathbf{Z}_T$ from Eq. 5 must be established. This is nontrivial and requires several novel techniques inspired by [11]. The details are in Appendix E.3.1. Theorem 2.2 gives the main guarantee for the spectral filtering algorithm, which states that Algorithm 3 instantiated with some choice of polynomial $p_n^c(\cdot)$ achieves regret

$$\tilde{O}\left(\left(n\|c\|_1 + T^{7/6}\max_{\alpha \in \mathbb{C}_\beta} |p_n^c(\alpha)|\right)(n+k)\sqrt{d_{\text{out}}}\sqrt{T}\right).$$

Both this theorem, as well as our new guarantee for convex regression, leads us to the following question: **Is there a universal choice of polynomial $p_n(x)$, where $n$ is independent of hidden dimension, which guarantees sublinear regret?**

## 3.1 Using the Chebyshev Polynomial over the Complex Plane

For the real line, the answer to this question is known to be positive using the Chebyshev polynomials of the first kind. In general, the $n^{\text{th}}$ (monic) Chebyshev polynomial $M_n(x)$ satisfies $\max_{x \in [-1,1]} |M_n(x)| \leq 2^{-(n-1)}$. However, we are interested in a more general question over the complex plane. Since we care about linear dynamical systems that evolve according to a general asymetric matrix, we need to extending our analysis to $\mathbb{C}_\beta$. This is a nontrivial extension since, in general, functions that are bounded on the real line can grow exponentially on the complex plane. Indeed, $2^{n-1} M_n(x) = \cos(n \arccos(x))$ and while $\cos(x)$ is bounded within $[-1, 1]$ for any $x \in \mathcal{R}$, over the complex numbers we have $\cos(z) = \frac{1}{2}(e^{iz} + e^{-iz})$, which is unbounded. Thus, we analyze the Chebyshev polynomial on the complex plane and provide the following bound.

**Lemma 3.1.** Let $z \in \mathbb{C}$ be some complex number with magnitude $|\alpha| \leq 1$. Let $M_n(\cdot)$ denote the $n$-th monic Chebyshev polynomial. If $|\arg(z)| \leq 1/64n^2$, then $|M_n(z)| \leq 1/2^{n-2}$.

We provide the proof in Appendix B. We also must analyze the magnitude of the coefficients of the Chebyshev polynomial, which can grow exponentially with $n$. We provide the following result.

**Lemma 3.2.** Let $M_n(\cdot)$ have coefficients $c_0, \ldots, c_n$. Then $\max_{k=0,\ldots,n} |c_k| \leq 2^{0.3n}$.

The proof of Lemma 3.2 is in Appendix B. Together, these two lemmas are the fundamental building block for universal sequence preconditioning and for obtaining our new regret bounds.

## 4 Experimental Evaluation

We empirically validate that convolutional preconditioning with Chebyshev or Legendre coefficients yields significant online regret improvements across various learning algorithms and data types. Below we summarize our data generation, algorithm variants, hyperparameter tuning, and evaluation metrics.

### 4.1 Synthetic Data Generation

We generate $N = 200$ sequences of length $T = 2000$ via three mechanisms: (i) a noisy linear dynamical system, (ii) a noisy nonlinear dynamical system, and (iii) a noisy deep RNN. Inputs $\mathbf{u}_{1:T} \sim \mathcal{N}(0, I)$.

**Linear Dynamical System.** Sample $(\mathbf{A}, \mathbf{B}, \mathbf{C})$ with $\mathbf{A} \in \mathcal{R}^{300 \times 300}$ having eigenvalues $\{z_j\}$ drawn uniformly in the complex plane subject to $\text{Im}(z_j) \leq \tau_{\text{thresh}}$ and $L \leq |z_j| \leq U$, and $\mathbf{B}, \mathbf{C} \in \mathcal{R}^{300}$. Then

$$\mathbf{x}_t = \mathbf{A}\,\mathbf{x}_{t-1} + \mathbf{B}\,\mathbf{u}_t, \quad \mathbf{y}_t = \mathbf{C}\,\mathbf{x}_t + \epsilon_t, \; \epsilon_t \sim \mathcal{N}(0, \sigma^2 I).$$

**Nonlinear Dynamical System.** Similarly sample $(\mathbf{A}_1, \mathbf{B}_1, \mathbf{C})$ and $(\mathbf{A}_2, \mathbf{B}_2)$ with $\mathbf{A}_i \in \mathcal{R}^{10 \times 10}$, $\mathbf{B}_i, \mathbf{C} \in \mathcal{R}^{10}$. Then

$$\mathbf{x}_t^{(0)} = \mathbf{A}_1 \mathbf{x}_{t-1} + \mathbf{B}_1 \mathbf{u}_t, \; \mathbf{x}_t^{(1)} = \sigma\big(\mathbf{x}_t^{(0)}\big), \; \mathbf{x}_t = A_2 \mathbf{x}_t^{(1)} + B_2 \mathbf{u}_t, \; \mathbf{y}_t = \mathbf{C}\mathbf{x}_t + \epsilon_t.$$

**Deep RNN.** We randomly initialize a sparse 10-layer stack of LSTMs with hidden dimension 100 and ReLU nonlinear activations. Given $\mathbf{u}_{1:T}$ we use this network to generate $\mathbf{y}_{1:T}$.

### 4.2 Algorithms and Preconditioning Variants

We evaluate the following methods: (1) **Regression** (Alg. 2) , (2) **Spectral Filtering** (Alg. 3) , (3) **DNN Predictor**: $n$-layer LSTM with dims $[d_1, \ldots, d_n]$, ReLU. Each method is applied with one of:

1. *Baseline:* no preconditioning
2. *Chebyshev:* $\mathbf{c}_{0:n}$ are the coefficients for the $n$th-Chebyshev polynomial. Note that when $n = 2$ we have $\mathbf{c}_0 = 1$ and $\mathbf{c}_1 = -1$ and therefore this is the method of *differencing* discussed in the introduction.
3. *Legendre:* $\mathbf{c}_{0:n}$ are the coefficients for the $n$th-Legendre polynomial

4. *Learned:* $c_{0:n}$ is a parameter learned jointly with the model parameters

We test polynomial degrees $n \in \{2, 5, 10, 20\}$. This choice of degrees shows a rough picture of the impact of $n$.

**Hyperparameter Tuning.** To ensure fair comparison, for each algorithm and conditioning $\mathbf{c}$ variant we perform a grid search over learning rates $\eta \in \{10^{-3}, 10^{-2}, 10^{-1}\}$, selecting the one minimizing average regret across the $N$ sequences. In the case of the learned coefficients, we sweep over the 9 pairs of learning rates $(\eta_{\text{model}}, \eta_{\text{coefficients}}) \in \{10^{-3}, 10^{-2}, 10^{-1}\} \times \{10^{-3}, 10^{-2}, 10^{-1}\}$.

## 4.3 Results

Tables 2–4 report the mean $\pm$ std of the absolute error over the final 200 predictions, averaged across 200 runs. In the linear and nonlinear cases we train a 2-layer DNN (dims $(64, 128)$); for RNN-generated data we match the 10-layer (100-dim) generator.

*Key observations:*

- Preconditioning drastically reduces baseline errors for all algorithms and data types.

- Chebyshev and Legendre yield nearly identical gains.

- For Chebyshev and Legendre, once the degree is higher than $5 - 10$ the performance degrades since $\|\mathbf{c}\|_1$ gets very large (see our Lemma 3.2 which shows that these coefficients grow exponentially fast).

- Improvements decay as the complex threshold $\tau_{\text{thresh}}$ increases, consistent with our theoretical results which must bound $\text{Im}(z_j)$.

- Learned coefficients excel with regression and spectral filtering but destabilize the DNN on nonlinear and RNN-generated data.

| Setting | Baseline | Chebyshev | | | Legendre | | | Learned | | | |
|---|---|---|---|---|---|---|---|---|---|---|---|
| | | Deg. 2 | Deg. 5 | Deg. 10 | Deg. 2 | Deg. 5 | Deg. 10 | Deg. 2 | Deg. 5 | Deg. 10 | Deg. 20 |
| **Regression** | | | | | | | | | | | |
| $\tau_{\text{thresh}} = 0.01$ | **0.74 ± 0.28** | 0.25 ± 0.09 | **0.15 ± 0.07** | 0.77 ± 0.31 | 0.36 ± 0.13 | **0.14 ± 0.06** | 0.64 ± 0.26 | 0.52 ± 0.19 | 0.27 ± 0.11 | **0.17 ± 0.07** | 0.24 ± 0.09 |
| $\tau_{\text{thresh}} = 0.1$ | **1.92 ± 0.81** | 0.84 ± 0.27 | **0.66 ± 0.18** | 1.90 ± 0.67 | 1.10 ± 0.40 | **0.63 ± 0.17** | 1.66 ± 0.58 | 1.34 ± 0.43 | 0.57 ± 0.14 | **0.55 ± 0.14** | 0.56 ± 0.14 s |
| $\tau_{\text{thresh}} = 0.9$ | **2.47 ± 0.89** | 1.59 ± 0.56 | 2.18 ± 0.79 | 2.68 ± 0.48 | **1.64 ± 0.58** | 1.94 ± 0.70 | 2.63 ± 0.45 | 1.73 ± 0.59 | 0.83 ± 0.25 | 0.68 ± 0.27 | **0.63 ± 0.26** |
| **Spectral Filtering** | | | | | | | | | | | |
| $\tau_{\text{thresh}} = 0.01$ | **5.94 ± 3.37** | 1.72 ± 0.95 | **0.69 ± 0.38** | 3.25 ± 1.79 | 2.78 ± 1.56 | **0.66 ± 0.36** | 2.74 ± 1.51 | 1.99 ± 0.94 | **0.54 ± 0.29** | 0.55 ± 0.25 | 0.61 ± 0.25 |
| $\tau_{\text{thresh}} = 0.1$ | **0.89 ± 0.34** | 0.42 ± 0.11 | **0.34 ± 0.07** | 0.86 ± 0.28 | 0.54 ± 0.17 | **0.33 ± 0.07** | 0.76 ± 0.24 | 0.69 ± 0.28 | **0.31 ± 0.06** | 0.37 ± 0.08 | 0.45 ± 0.28 |
| $\tau_{\text{thresh}} = 0.9$ | 10.17 ± 8.80 | 9.87 ± 8.90 | 12.66 ± 8.18 | 32.90 ± 22.02 | **9.42 ± 8.39** | 11.53 ± 7.46 | 28.83 ± 19.24 | 7.93 ± 4.42 | 6.31 ± 4.19 | **5.73 ± 3.80** | 5.86 ± 4.02 |
| **2-layer DNN** | | | | | | | | | | | |
| $\tau_{\text{thresh}} = 0.01$ | 4.49 ± 2.02 | **2.31 ± 1.18** | 2.62 ± 1.52 | 10.36 ± 6.05 | 2.79 ± 1.34 | **2.35 ± 1.35** | 8.92 ± 5.20 | 2.89 ± 1.24 | 1.56 ± 0.64 | 0.79 ± 0.25 | **0.40 ± 0.16** |
| $\tau_{\text{thresh}} = 0.1$ | 9.41 ± 7.34 | 2.66 ± 1.76 | **1.52 ± 0.72** | 4.65 ± 3.03 | 4.24 ± 3.06 | **1.44 ± 0.71** | 4.03 ± 2.64 | 6.54 ± 4.32 | 3.22 ± 1.93 | 1.59 ± 0.95 | **0.80 ± 0.48** |
| $\tau_{\text{thresh}} = 0.9$ | 2.45 ± 1.31 | **2.24 ± 1.34** | 3.49 ± 2.27 | 11.48 ± 8.15 | **2.17 ± 1.25** | 3.10 ± 2.00 | 9.97 ± 7.05 | 1.29 ± 0.66 | 0.71 ± 0.34 | 0.43 ± 0.18 | **0.24 ± 0.11** |

Table 2: Linear dynamical system data (detailed in Sec. 4.1) across varying complex threshold $\tau_{\text{thres}}$.

| Setting | Baseline | Chebyshev | | | Legendre | | | Learned | | | |
|---|---|---|---|---|---|---|---|---|---|---|---|
| | | Deg. 2 | Deg. 5 | Deg. 10 | Deg. 2 | Deg. 5 | Deg. 10 | Deg. 2 | Deg. 5 | Deg. 10 | Deg. 20 |
| **Spectral Filtering** | | | | | | | | | | | |
| $\tau_{\text{thresh}} = 0.01$ | **153.8 ± 15.7** | 43.7 ± 19.4 | 0.92 ± 0.31 | **0.26 ± 0.15** | 78.4 ± 34.7 | 2.82 ± 1.17 | **0.34 ± 0.19** | 1.04 ± 0.29 | 0.07 ± 0.03 | **0.05 ± 0.03** | 0.07 ± 0.03 |
| $\tau_{\text{thresh}} = 0.01$ | **124.1 ± 68.4** | 33.4 ± 17.3 | 2.02 ± 1.23 | **0.36 ± 0.31** | 56.7 ± 30.5 | 3.81 ± 1.67 | **0.35 ± 0.26** | 2.85 ± 1.13 | 0.04 ± 0.01 | **0.02 ± 0.01** | 0.07 ± 0.02 |
| $\tau_{\text{thresh}} = 0.9$ | **165.5 ± 84.5** | 43.41 ± 16.39 | **1.27 ± 0.37** | 1.90 ± 0.80 | 76.9 ± 29.1 | 3.32 ± 1.09 | **1.64 ± 0.69** | 1.79 ± 0.61 | **0.10 ± 0.04** | 0.12 ± 0.05 | 0.17 ± 0.07 |
| **2-layer DNN** | | | | | | | | | | | |
| $\tau_{\text{thresh}} = 0.01$ | 10.46 ± 5.10 | 3.45 ± 1.19 | **0.19 ± 0.16** | 0.43 ± 0.22 | 3.55 ± 1.56 | **0.18 ± 0.14** | 0.37 ± 0.19 | 45.40 ± 11.35 | 25.34 ± 9.06 | 12.73 ± 4.60 | **6.43 ± 2.32** |
| $\tau_{\text{thresh}} = 0.1$ | 4.20 ± 1.10 | 3.38 ± 0.79 | **0.14 ± 0.05** | 0.41 ± 0.16 | 3.42 ± 0.79 | 0.25 ± 0.14 | 0.36 ± 0.13 | 67.70 ± 28.40 | 31.99 ± 11.78 | 15.95 ± 5.87 | **8.01 ± 2.95** |
| $\tau_{\text{thresh}} = 0.9$ | 6.72 ± 2.35 | 2.45 ± 1.12 | **0.08 ± 0.03** | 0.28 ± 0.15 | 3.35 ± 0.98 | **0.20 ± 0.07** | 0.24 ± 0.13 | 58.65 ± 17.09 | 28.74 ± 8.07 | 14.39 ± 4.06 | **7.26 ± 2.05** |

Table 3: Nonlinear data (detailed in Sec. 4.1) across varying complex threshold $\tau_{\text{thres}}$.

| Setting | Baseline | Chebyshev | | | Legendre | | | Learned | | | |
|---|---|---|---|---|---|---|---|---|---|---|---|
| | | Deg. 2 | Deg. 5 | Deg. 10 | Deg. 2 | Deg. 5 | Deg. 10 | Deg. 2 | Deg. 5 | Deg. 10 | Deg. 20 |
| 10-layer DNN | **0.54 ± 0.23** | 0.29 ± 0.10 | **0.08± 0.03** | 0.13 ± 0.05 | 0.37 ± 0.14 | **0.09± 0.03** | 0.12± 0.04 | 1.49± 0.93 | 2.13± 1.05 | 1.04± 0.51 | **0.5 ± 0.24** |

Table 4: Performance (average absolute error of the last 200 predictions) of a 10-layer DNN (detailed in Sec. 4.2) on data generated from the same model (detailed in Sec. 4.1).

## 4.4 ETTh1 Dataset

To evaluate whether our proposed preconditioning approach generalizes to real-world time series, we conduct experiments on the well-established **ETTh1** dataset from the Electricity Transformer Temperature (ETT) benchmark [37]. The ETTh1 dataset consists of continuous hourly measurements of load and oil temperature collected from electricity transformers and has been used in several recent works [37, 35, 32, 22, 23, 36, 31]. We study the effect of preconditioning on a 10-layer LSTM with hidden dimension 100 per layer using the Adam optimizer. We set the horizon to be $T = 5000$ and we sweep over a broader range of learning rates $\eta \in \{10^{-j}\}_{j=0,1,2,3,4,5}$. As before we consider (i) no preconditioning (baseline), (ii) fixed Chebyshev coefficients, (iii) fixed Legendre coefficients, and (iv) coefficients learned jointly with model parameters. As seen in Figure 1, preconditioning with Chebyshev and Legendre for degree 5 the best performance after only the first 1000 iterations, while the performance of jointly learning the coefficients is worse at this stage. The performance of all three preconditioning methods are roughly on par with each other by 2500 iterations and by the full horizon $T = 5000$, jointly learning the coefficients results in the best average prediction error.

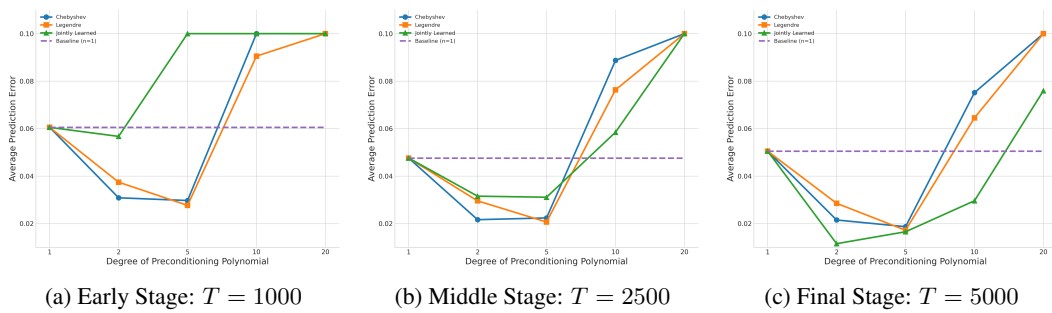

(a) Early Stage: $T = 1000$    (b) Middle Stage: $T = 2500$    (c) Final Stage: $T = 5000$

Figure 1: Absolute prediction error on final 200 predictions averaged over 10 independent runs for 10-layer LSTM with layer dimension 100 using Adam optimizer and sweeping over learning rates for each run.

## 5 Discussion

There are many settings in machine learning where universal, rather than learned, rules have proven very efficient. For example, physical laws of motion can be learned directly from observation data. However, Newton's laws of motion succinctly crystallize very general phenomenon, and have proven very useful for large scale physics simulation engines. Similarly, in the theory of mathematical optimization, adaptive gradient methods have revolutionized deep learning. Their derivation as a consequence of regularization in online regret minimization is particularly simple [19], and thousands of research papers have not dramatically improved the initial basic ideas. These optimizers are, at the very least, a great way to initialize learned optimizers [34].

By analogy, our thesis in this paper is that universal preconditioning based on the solid theory of dynamical systems can be applicable to many domains or, at the very least, an initialization for other learning methods.

## Acknowledgments and Disclosure of Funding

Elad Hazan gratefully acknowledges funding from the Office of Naval Research and Open Philanthropy.

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
