# OpenReview forum: "Universal Sequence Preconditioning"
_NeurIPS.cc/2025/Conference — NeurIPS 2025 spotlight_

### Official Review · Reviewer_S8cH · 2025-06-27

**Clarity:** 3
**Significance:** 2
**Originality:** 3
**Rating:** 4
**Confidence:** 4

**Summary:**

This paper aims to solve sequential prediction via preconditioning. Based on Chebyshev polynomials, the authors propose a universal sequence preconditioning methodology that be applied as a convolutional preconditioned regression and convolutional preconditioned spectral filtering. The authors prove the convergence rate for a particular linear dynamic system, and experiment the methods on various settings, including linear dynamic system, non-linear dynamic system, and deep RNN.

**Questions:**

The questions follow from the "Weakness":
1. Could you fix all the missing references, such as "Eq.TO DO" and "Algorithm 4"?
2. Could you clarify all the implicit dependence?
3. Could you discuss how sharp your theoretical results are?
4. Could you provide more analysis comparing Chebyshev and Legendre polynomials in terms of their strength and weakness in terms of theoretical guarantee and empirical performance?

**Ethical Concerns:**

["NO or VERY MINOR ethics concerns only"]

**Final Justification:**

The authors adequately address my concerns and discuss the possible improvements on the bound for Chebyshev polynomials. I agree with other reviewers' that experiments beyond synthetic data will strengthen the paper.

**Limitations:**

Yes.

**Paper Formatting Concerns:**

I do not notice any major formatting issue.

**Quality:**

3

**Strengths And Weaknesses:**

Strength:

1. The authors prove the explicit convergence rate.
2. The authors give intuition behind the proof and the trade-off with respect to parameters (such as the degree of polynomials and the threshold).
3. The algorithms are tested empirically.

Weakness:
1. There is unclear implicit dependence: Theorem 2.1 does not seem to be dependent on the degree of the polynomial $n$, which is surprising considered Algorithm 2 heavily depends on the choice of $n$. I would suspect that there is an implicit dependence on $n$ hidden somewhere, for example perhaps like the scaling $n=\log(T)$ on line 114? If the result is completely independent of $n$, then the authors should explain why that is the case.
2. Slow convergence rate: The convergence rate in Theorem 2.1 is $T^{-2/13}$ (up to log factors), which is quite slow. It is unclear from this paper whether this is the actual empirical convergence rate, or a pessimistic bound due to proof technique.

---

> ### Author Rebuttal · Authors · 2025-07-29
>
> We thank the reviewer for their thougthful review!
>
> Weakness 1) Thank you so much for finding this. We implicitly choose n to optimally depend on T (line 702 on page 23) and did not make this choice clear in the statement of the theorem. We will change this. If the reviewer is curious, the dependence on degree n is in line 701 and we could include this and discuss it in the main body of the paper if the reviewer thinks it would be a good idea.
>
> Weakness 2) The reason the rate looks slow (i.e. not the typical $O(\sqrt{T})$)  is that it is independent of hidden dimension and extends to some systems with asymmetric transition matrices. It is unclear if you can hope for better in the worst case (i.e. letting hidden dimension grow very large and the transition matrix having large imaginary component) which we discuss in Section A.3. Importantly, the method shows huge convergence gains empirically as our synthetic experiments show.
>
> Questions
> 1) Yes, we apologize for the typos and thank the author for seeing the merit of the work despite typos. A sudden illness caused the writing of the main body of the paper to be rushed.
>
>
> 2) We clarified the implicit dependence above that in the case where $p_n(x)$ is chosen to be the n-th degree monic chebyshev polynomial, we pick $n = (10/13) \log_2 (8 T^{3/2}/ 3 \sqrt{d})$. We note that the proof of the theorem starting in line 699 makes all the dependencies explicit. We will include the choice of n in the statement of the theorem and make this clearer. We thank the reviewer for noting it!
>
>
> 3) We discuss sharpness in Section A.3 in terms of avoiding polynomial/linear dependence on the hidden dimension of the transition matrix. Please let us know if you have further questions on this and we would be very happy to answer!
>
>
> 4) Only Chebyshev polynomials provably give us sublinear regret right now. Our Lemma 4.1 holds for Chebyshev polynomials and we note that a similar theorem (at least on the region [-1,1] we aren’t sure about extending it to the complex plane) for Legendre polynomials exists but is a factor $\sqrt{\pi n}$ larger. On the other hand, the coefficients of the Chebyshev polynomial grow slightly faster than the coefficients of the Legendre polynomial. We prove that the maximum absolute value coefficient for the n-th monic Chebyshev can be bounded by $2^0.3n$ and I believe you could show something slightly tighter (ie. $2^{0.28n}$) for Legendre but it isn’t as obvious how to do it. Empirically we don’t see a huge difference between the two but it might be interesting to compare them on more applications in the future.

---

> > ### Comment · Reviewer_S8cH · 2025-08-05
> >
> > Thank you for the response. I appreciate that you adequately address my concerns and discuss the possible improvements on the bound for Chebyshev polynomials. Although I agree with other reviewers' that experiments beyond synthetic data will strengthen the paper, I acknowledge the theoretical contribution of this paper.

---

### Official Review · Reviewer_EHYC · 2025-06-30

**Clarity:** 2
**Significance:** 3
**Originality:** 3
**Rating:** 5
**Confidence:** 3

**Summary:**

In this paper, the authors investigate which input sequence modification method would facilitate the learning process of model learning and improve its prediction accuracy, which is referred as the preconditioning problem. Inspired by the theory of linear dynamic systems, they propose a novel and universal preconditioning method that convolves the input sequence with the coefficients of the Chebyshev or Legendre polynomials. They further combine the proposed universal preconditioning method with the spectral filtering method and propose a new spectral filtering algorithm. The authors then provide comprehensive theoretic analysis on the proposed preconditioning method and shown that its regret bound is smaller than that of previous methods. Particularly, the proposed spectral filtering algorithm has a sublinear hidden-dimensional free regret bound, even if the hidden transition matrix is asymmetric. Experiments on synthetic data show that the proposed preconditioning method is applicable to diverse data types and learning algorithms.

**Questions:**

1. There is no explicit definition of $p _n(\cdot)$ in the last equation presented on page 5. Its definition only appears in line 2 of algorithm 3, that is, $p _n(x) = \sum _{i=0}^n c_i x^{n-i}$. It is encouraged to add this definition before the last equation on page 5 to enhance clarity of expression.
2. In line 670, the correct one should be $G \leq \sqrt{d _{\rm out}}$, according to the inequality between line 669 and 670.
3. In the equation between line 664 and 665, the authors show that $\sum _{t=1}^T \ell _t(\\mathbf{Q}^t) - \min _{\\mathbf{Q}^{\star} \in \\mathcal{K}} \sum _{t=1}^T \ell_t (\\mathbf{Q}^{\star}) \leq \frac{3}{2} GD \sqrt{T}$. Also, they show that $D \leq 2 C _{\rm domain} n^2 \Vert \\mathbf{c} \Vert_1 $ and $G \leq \sqrt{d _{\rm out}}$ in line 665 and line 670, respectively. Thus, the correct one should be $\sum _{t=1}^T \ell _t(\\mathbf{Q}^t) - \min _{\\mathbf{Q}^{\star} \in \\mathcal{K}} \sum _{t=1}^T \ell_t (\\mathbf{Q}^{\star}) \leq 3 C _{\rm domain} n^2 \sqrt{d _{\rm out}} \Vert \\mathbf{c} \Vert _1$ in line 671.
4. In line 682, the authors show that $\Vert \hat{\\mathbf{y}} _t - \\mathbf{y} _t \Vert_1 = \Vert \sum _{s=0}^{t-n-1} \\mathbf{C} p^{\\mathbf{c}} _n(A) \\mathbf{A}^s \\mathbf{B} \\mathbf{u} _{t-n-s} \Vert _1$ holds. Then, they further show that $\Vert \\mathbf{C} p^{\\mathbf{c}} _n(A) \\mathbf{A}^s \\mathbf{B} \Vert _1 \leq \max _{\\lambda \in \\lambda(\\mathbf{A})} \vert p^{\\mathbf{c}} _n(\\lambda) \vert \cdot \Vert \\mathbf{C} \Vert \Vert \\mathbf{B} \Vert$ holds for any $j = 0, \ldots, t-n-1$. Since $\Vert \\mathbf{u} _j \Vert _1 \leq 1$, by triangleq inequality we should have
$$
\begin{aligned}
\Vert \sum _{s=0}^{t-n-1} \\mathbf{C} p^{\\mathbf{c}} _n(A) \\mathbf{A}^s \\mathbf{B} \\mathbf{u} _{t-n-s} \Vert _1 \leq & \sum _{s=0}^{t-n-1} \Vert \\mathbf{C} p^{\\mathbf{c}} _n(A) \\mathbf{A}^s \\mathbf{B} \\mathbf{u} _{t-n-s} \Vert _1 \leq \sum _{s=0}^{t-n-1} \Vert \\mathbf{C} p^{\\mathbf{c}} _n(A) \\mathbf{A}^s \\mathbf{B} \Vert _1 \Vert \\mathbf{u} _{t-n-s} \Vert _1 \leq \sum _{s=0}^{t-n-1} \Vert \\mathbf{C} p^{\\mathbf{c}} _n(A) \\mathbf{A}^s \\mathbf{B} \Vert _1 \\\\
& \leq \sum _{s=0}^{t-n-1} \max _{\\lambda \in \\lambda(\\mathbf{A})} \vert p^{\\mathbf{c}} _n(\\lambda) \vert \cdot \Vert \\mathbf{C} \Vert \Vert \\mathbf{B} \Vert = (t-n-1) \max _{\\lambda \in \\lambda(\\mathbf{A})} \vert p^{\\mathbf{c}} _n(\\lambda) \vert \cdot \Vert \\mathbf{C} \Vert \Vert \\mathbf{B} \Vert.
\end{aligned}
$$
Therefore, we should have
$$
\min _{\\mathbf{Q}^{\star} \in \\mathcal{K}} \sum _{t=1}^T \ell_t (\\mathbf{Q}^{\star}) \leq \sum _{t=1}^T \ell _t (\hat{\\mathbf{Q}}) = \sum _{t=1}^T \Vert \hat{\\mathbf{y}} _t - \\mathbf{y} _t \Vert_1 \leq \sum _{t=1}^T (t-n-1) \max _{\\lambda \in \\lambda(\\mathbf{A})} \vert p^{\\mathbf{c}} _n(\\lambda) \vert \cdot \Vert \\mathbf{C} \Vert \Vert \\mathbf{B} \Vert.
$$
It seems that it is possible to derive a slightly sharper bound, that is, $\frac{(1+T)T}{2}$. It is also encouraged to include the above details in the paper enhance clarity of expression.
5. In Eq. (18), the correct one seems to be $\Vert U _{n-1}(z) \Vert \leq n$. If $n$ is even, we have $\Vert U _{n-1}(z) \Vert \leq 2 \sum _{j \geq 0, j even}^n T_j (z) \leq 2 \cdot \frac{n}{2} \cdot 2 = 2n$, using Eq. (16) that $\Vert T_j(z) \Vert \leq 2$ for any $j$. The case that $n$ is    add is similar.
6. I am confused about the third line of the equation in line 918. It seems that we can only get
$$
\sum _{k=0}^\infty a _k \vert r \vert^{2k+1} \min (1,(2k+1)\vert \theta \vert) = \sum _{k=0}^\infty a _k \vert r \vert^{2k} \cdot \vert r \vert \min (1,(2k+1)\vert \theta \vert) \leq \sum _{k=0}^\infty a _k \vert r \vert \min (1,2(2k+1)\vert \sin(\theta) \vert) = \sum _{k=0}^\infty a _k \min (\vert r \vert,2(2k+1)\vert r \sin(\theta) \vert).
$$
Could the authors provide more explanations on this?
7. The polynomials used in this paper include Chebyshev or Legendre polynomials, and both of them are orthogonal polynomials. Since Chebyshev or Legendre polynomials are special cases of Jacobi polynomial, is it possible to directly use the Jacobi polynomial ?

**Ethical Concerns:**

["NO or VERY MINOR ethics concerns only"]

**Final Justification:**

My previous concerns about the paper were mainly regarding some mathematical formulas, which the author has explained quite clearly in their rebuttal.

**Limitations:**

The authors have addressed the limitations of their work in line 278-380.

**Paper Formatting Concerns:**

I do not find any formatting issues in this paper.

**Quality:**

3

**Strengths And Weaknesses:**

**Strengths**
- The motivation of this work is clear and theoretic solid. The inspiration comes from the theory of linear dynamic systems, and the authors provide comprehensive theoretic analysis for the proposed preconditioning method and show that it can achieve smaller regret bound.
- The experiments on synthetic data verify that the proposed method has universal adaptability. It can be applied on data that do not form a linear dynamic system and various learning algorithms including recurrent neural network.

**Weakness**
- The major weakness of this paper is the presentation quality. The manuscript has several typographical errors, insufficient contextual details in certain parts, and requires improvement in logical coherence. For example, sentences ''...potentially shrinking the domain'' in line 33 and ''..., is a hard problem'' in line 68 are missing a period. Besides, the derivations of some theoretical results exhibit ambiguities that require clarification. See Question part for more detail.
- The experiments are conducted only on synthetic data without validation on real-world data, which makes the proposed method less convincing. It is encouraged to conduct experiments on real-world datasets.

---

> ### Author Rebuttal · Authors · 2025-07-29
>
> We thank the reviewer for their thougthful review! First we respond to the two weaknesses and then the questions.
>
> Weakness 1)  General response to the clarity/typos: The reviewer is correct, and we apologize for the hasty writing at times. A main author got very sick and it caused the main body writing to be rushed before submission. We appreciate the reviewer’s reading of the supplementary material and seeing through the unpolished writing. For most of the ambiguities in the technical details-- the difference is just that we were not concerned with optimizing constants. But we deeply value the reviewer’s time and careful questions and we will make the suggested improvements!
>
> Weakness 2) This is a fair point, we are looking into this! We have done some experiments on LLMs showing the benefit of the method and would be happy to add a small model over FineWeb.edu 10B in the final version. Please note though that the paper is mostly theoretical, and is pretty involved at that.  We believe that the paper is of interest even without more experiments, although it does propose a practical method as the reviewer notes.
>
>
> Questions 1-4) We thank the reviewer for their incredible/careful attention and time to read through these details. We note that we were not concerned about optimizing the constants for the bound, but it is always a good idea to do so and we will make these edits.
>
> Question 5) We appreciate the reviewer finding a missing factor of 2 and we will propagate this change.
>
> Question 6) Your equation is upper bounded by our line 3 since $|r| \leq 1$ so you can replace the LHS of the minimum with 1 and then observe that for any $a,b \geq 0$, $\min (a,2b) \leq 2 \min(a,b)$. Let us know if you would like further explanation!
>
> Question 7) It’s a good idea! However, for the theory, we need the property that 1) the maximum abosolute value of the polynomial over the domain (which represents the eigenvalues of the system transition matrix) be very small and that 2) the coefficients of the polynomial also stay somewhat small. Of the shelf, Jacobi polynomials don’t guarantee this property.

---

> > ### Comment · Reviewer_EHYC · 2025-08-05
> > **Official Comment by Reviewer EHYC**
> >
> > Thanks the authors for their responses, which have adequately addressed my concerns. I encourage the authors to carefully check all the formulas and corresponding proofs in the paper to make it more rigorous. Overall, I think this is an interesting work, and I have raised my score to 5.

---

### Official Review · Reviewer_2AEH · 2025-07-02

**Clarity:** 1
**Significance:** 3
**Originality:** 3
**Rating:** 4
**Confidence:** 2

**Summary:**

The paper investigates preconditioning of sequences (of inputs and outputs to linear dynamical systems) for the purpose of improving prediction accuracy. The paper proposes a preconditioning method that convolves the "input sequence" (actually the "output sequence" according to Algorithm 1) with the coefficients of the Chebyshev polynomial. The paper proves that this "universal sequence preconditioning" can improve the regret of two popular prediction algorithms: regression (in which the predictor coefficients are updated online) and (a new variant) of spectral filtering. Numerical examples/experiments are run on synthetic data, coming from (noisy) linear dynamical systems, and nonlinear dynamical systems (including a neural network).

**Questions:**

Can you please provide a clear and precise description of the problem that this paper addresses? In particular, is T assumed to be known in advance? (It seems so). Can you please comment on this assumption (in many applications, predictors are required to run for arbitrarily long T).

Can you please explain the nonlinear dynamical system in line 220 more clearly?

**Ethical Concerns:**

["NO or VERY MINOR ethics concerns only"]

**Final Justification:**

The authors have addressed my concerns regarding the (lack of) clarity in the original manuscript; I am confident that the revised manuscript will be much improved. I have increased my score accordingly.

**Limitations:**

Yes

**Paper Formatting Concerns:**

None.

**Quality:**

3

**Strengths And Weaknesses:**

The main strengths of the work are:
- importance of problem area (investigation of preconditioning strategies for prediction)
- mathematical rigor: I did not check all the proofs carefully, but the technical appendices were generally quite well-written and easy to follow.
- the inclusion of numerical experiments is commendable for a paper that mostly makes theoretical contributions.

In my view, the main weaknesses of the work are:
- Clarity of the presentation in the main paper. In contrast to the technical appendices in the supplementary material, the main manuscript is quite poorly written. For example, Line 92 refers to "Eq: TODO" (although this equation is given in the supplementary materials). Also, in Section 2.3 the regret of the spectral filtering algorithm is given as $\tilde{O}(\sqrt{T})$ whereas in the supplementary material it is given as $\tilde{O}(T^{2/3})$.
- Overall, the quality of the supplementary materials seems higher than that of the main submitted manuscript. (If I based my review on the main manuscript alone, I would probably recommend rejection).
- Clarity: the paper also misses a clear problem statement. Readers familiar with the general problem setting can probably infer the overall goals and assumptions, but in my opinion the problem that the paper addresses should be precisely stated. For example, in the introduction the authors write "The focus of this paper are not methods for sequence prediction, but rather preconditioning algorithms", yet it appears that one of the main contributions of the paper is a more broadly applicable spectral filtering algorithm? Also, it's not immediately obvious whether the authors consider online or offline learning. For example, in Algorithm 1, line 7 makes it clear that prediction algorithms are learned offline, based on training data. However, algorithm 2 and 3 appear to update the prediction algorithm online. (Most likely the "learning" of the prediction algorithm in line 7 of Algorithm 1 refers to the tuning of hyper-parameters, and actual prediction algorithm is updated online -- still, it would be good to make this clear).
- Clarity: overall, the main manuscript seems quite hastily written, e.g. "Eqn: TODO", sometime they refer to their algorithm as UPS instead of USP (e.g. Table 1).
- Quality: although the core technical developments seem fairly sound, some of the statements around peripheral aspects are questionable. For example, on line 121, they claim that system identification is difficult. However, in their setting (linear systems with no noise -- which is the setting described in the supplementary material) system identification is easy (e.g. any subspace method should recover the system parameters perfectly). Also, there seems to be some confusion around "closed-loop" components. On line 127 they make clear that by "closed-loop" they refer to autoregressive terms (e.g. functions of previous outputs), whereas on line 108 they refer to Q as a closed-loop component, even though Algorithm 3: Line 5 shows that Q combines only with past inputs, not past outputs.

Overall, the supplementary material seems relatively sound, but the sloppiness and lack of clarity in the main manuscript casts doubt on the quality of the work.

---

> ### Author Rebuttal · Authors · 2025-07-29
>
> We thank the reviewer for their thoughtful review! First we respond to some of the items mentioned regarding clarity:
>
>
> 1) Regret says $\sqrt{T}$ in Section 2.3 versus $T^{⅔}$ in the supplementary material: In lines 108-112 of the supplementary material (Section 2.3) we explain that r (the bound on the coefficients of the preconditioning method which appears in front of the sqrt(T)) can be chosen to depend on T. Indeed, for Chebyshev preconditioning we choose the degree of the polynomial to depend logarithmically on T and since we show the maximum norm of the coefficients grows with T, we have a slight dependence which is why you see a regret $T^{⅔}$. If this is unnecessarily confusing we are happy to write it differently!
>
>
> 2) Response to “For example, in the introduction the authors write "The focus of this paper are not methods for sequence prediction, but rather preconditioning algorithms", yet it appears that one of the main contributions of the paper is a more broadly applicable spectral filtering algorithm?”
> First, we apply preconditioning to both 1) spectral filtering and 2) convex regression and prove that it helps both algorithms. Moreover, in the experimental section, we apply preconditioning to a DNN algorithm. That is why we say that the goal is about preconditioning the sequence rather than focusing on one particular learning algorithm.
>
>
> 3) Response to offline vs online: In line 41-42 we say “Here we provide the offline train-test version of the method and give the online version in Appendix B as Algorithm 4.” Let us know if you would prefer just observing the online version of the algorithm for clarity, we are happy to remove the offline version or put it in Appendix B instead. All of our theorems are clear on which algorithm they refer to (and they are online just fyi).
> The learning of the prediction algorithm is typically done through gradient descent on the algorithm’s parameters. The prediction algorithm is not updated online– at test time you use the learned parameters of the algorithm. We are happy to remove Algorithm 1 and replace it with Algorithm 4– it is just there for further clarity for the reader.
>
>
> 4) Response to “For example, on line 121, they claim that system identification is difficult”: Thank you for catching this, we didn’t mean it quite as it is as written and will fix this. It’s true that there are many methods for system identification that we don’t mention, for example [2]. What we meant to highlight was that these methods have regret bounds that depend on hidden dimension, which can be prohibitively large whereas the method introduced in our work allows for a regret that depends only logarithmically on hidden dimension.
> [2] Max Simchowitz, Horia Mania, Stephen Tu, Michael I. Jordan, Benjamin Recht Proceedings of the 31st Conference On Learning Theory, PMLR 75:439-473, 2018.
>
>
> 5) General response to the clarity/typos: The reviewer is correct, and we apologize for the hasty writing at times. A main author got very sick and it caused the main body writing to be rushed before submission. We appreciate the reviewer’s reading of the supplementary material and seeing through the unpolished writing. Importantly, we hope the reviewer sees that our technical results and experiments were not rushed and are very sound. We promise to fix typos and polish the writing.
>
>
> Question 1) We would like to make sure we fully answer the reviewer’s concern regarding the problem statement. Can the reviewer please look at the statement of Theorem C.1 on line 658 and Theorem 2.1 on line 693 and clarify which aspects of the problem they would like to be better fleshed out? With respect to the horizon being known in advance– for spectral filtering the horizon T must be set in advance. T can be arbitrarily large but it does need to be known beforehand. This is a common assumption in online learning. Indeed, Hedge/EXP3/OGD all require knowing T in advance to tune the learning rate appropriately.
>
>
> Question 2) Yes. First, $\sigma$ represents the sigmoid function which we apply elementwise (we will make sure to define this in the text). Next, we sample A_1 and A_2 using the same as for the linear dynamical system. So, to be clear,  given a complex threshold $\tau_{\text{threshold}}$ we sample eigenvalues to be drawn uniformly at random to have radius between L and U, which are 0.9 and 0.9999 respectively, and complex part bounded by $\tau_{\text{threshold}}$. We then find an asymmetric real valued matrix with this spectrum. We generate B_1, B_2, and C randomly– each entry is an independent Gaussian– and then we renormalize them.

---

> > ### Comment · Reviewer_2AEH · 2025-08-06
> >
> > I would like to thank the authors for their detailed response.
> >
> > 1. Thank you for clearing this up. I think it's fine as it is; I probably just did not read the paper carefully enough the first time.
> >
> > 2. Thanks, this is also clear.
> >
> > 3. Thanks again. I guess clarity is always subjective, but given that both Algorithm 2 and Algorithm 3 are instantiations of the (online) Algorithm 4, one might argue that it would make more sense to put Algorithm 4 in the main body of the paper (possibly in place of, or at least in addition to, Algorithm 1)?
> >
> > 4. Thank you; I think the paper would benefit from a slightly more precise critique of system identification (as you have described in the rebuttal).
> >
> > 5. Thank you for this clarification. Given that the supplementary materials were of high quality, I have no doubt that the quality of the final revised manuscript will be greatly improved. (And I wish the author a very swift recovery).
> >
> > Q1: Regarding the problem statement: I probably did not express my question very clearly, but my general impression is that Section 1 appears to be quite general (in terms of applicability -- though I do note that linear dynamical systems are explicitly mentioned in 1.1.1), but then Theorem 2.1 concerns only linear dynamical systems, and Section 2.2 lacks a theorem at all. This is not a comment on the strength of the results (which are good), but rather a comment on the organization of the paper. Given that Theorem 2.1 and Theorem D.1 both apply to linear dynamical systems, might it make sense to make this setting -- namely, online prediction for linear dynamical systems --  explicit (perhaps in at the end of Section 1 or the beginning of Section 2)? As a concrete example, the reader does not even understand what noise assumptions are made (regarding the data generating process) without checking the appendices (and even then, the appendices say "we ignore dynamics and observation noise for clarity"?). A nice clean statement of the assumptions on the underlying data generating process in the body of the paper (instead of just Section A.1) might be useful to the reader?
> >
> > Q2: Thanks for the clarification.
> >
> > I am willing to increase my score.

---

> > > ### Author Response · Authors · 2025-08-07
> > > **Response to Q1**
> > >
> > > Q1: That makes a lot of sense, thank you for clarifying. Yes- we completely agree that it would be better for the reader if we introduce linear dynamical systems at the beginning of Section 2 (somewhat like what is written in section A.1 but with the standard noise assumption included) and clarify all terms and assumptions before the theorems. That way, the statements of the theorems are better contextualized. Thank you for this suggestion!

---

### Official Review · Reviewer_jQGL · 2025-07-03

**Clarity:** 3
**Significance:** 3
**Originality:** 4
**Rating:** 5
**Confidence:** 4

**Summary:**

This paper proposes a method to precondition time series data such as to improve sequential prediction. The main idea relies on taking the convolution of the input sequence with Chebyshev polynomial coefficients. The authors then show that this methodology improves the regret of different sequence prediction models, thereby providing strong theoretical motivations for their approach.

The authors further show that their methodology can be applied to multiple algorithms such as recurrent neural networks. And they demonstrate the improved performance of their preconditioning method on synthetic data.

**Questions:**

- Could the author include a non-synthetic dataset to their experiments to validate their approach in a less controlled setup ?
- The paper takes inspiration from differencing (subtracting each observation with the last one). However, the authors do not compare against this method in the experiments. Could the authors include that comparison ?

**Ethical Concerns:**

["NO or VERY MINOR ethics concerns only"]

**Limitations:**

yes

**Quality:**

3

**Strengths And Weaknesses:**

- This paper proposes a very creative approach to improving learning of sequences.
- The authors provide strong theoretical motivation and guarantees for their preconditioning approach
- The experimental results show significant improvement of their method over no preconditioning.
- However, the experimental section is limited to synthetic data alone, which is a pity given the trove of sequence data available. I would encourage the authors to include real-world data experiments to further strengthen the paper.

---

> ### Author Rebuttal · Authors · 2025-07-29
>
> We thank the reviewer for their thoughtful review!
>
>
> Question/weakness regarding nonsynthetic experiments) This is a fair point, we are looking into this! We have done some experiments on LLMs showing the benefit of the method and would be happy to add a small model over FineWeb.edu 10B in the final version. Please note though that the paper is mostly theoretical, and is pretty involved at that. We believe that the paper is of interest even without more experiments, although it does propose a practical method as the reviewer notes.
>
>
> Question on comparing to differencing) This method is actually the same as our method when n_coeffs=2, since the first Chebyshev polynomial is [1 -1], so we do compare to it. We will make it clear in the paper!

---

> > ### Comment · Reviewer_jQGL · 2025-08-01
> > **Answer to authors**
> >
> > Thank you for your answers.
> >
> > I still believe a small non-synthetic experiment would strengthen the paper.
> >
> > Thank you for clarifying in the paper that you indeed compare to differencing.

---

### Official Review · Reviewer_u5xs · 2025-07-04

**Clarity:** 3
**Significance:** 3
**Originality:** 3
**Rating:** 4
**Confidence:** 3

**Summary:**

This paper investigates the problem of preconditioning for sequential prediction, introducing a universal sequence preconditioning (USP) method grounded in the application of orthogonal polynomial filters (such as Chebyshev and Legendre polynomials) to input sequences. The authors provide a theoretical analysis demonstrating improvement in regret bounds for learning in linear dynamical systems, even in cases with asymmetric and marginally stable transition matrices. The paper applies USP to both convex regression and spectral filtering techniques, with rigorous regret analyses and empirical evaluation on synthetic data generated from linear, nonlinear, and deep RNN-based dynamical systems. Experimental results suggest that polynomial-based preconditioning yields consistent performance improvements across a range of models and data types.

**Questions:**

1. How the eigenvalue imaginary-part assumption for the transition matrix manifests in commonly encountered real-world time series?

2. What are the main computational bottlenecks?

3. Why analytic polynomial coefficients (Chebyshev/Legendre) are more robust than learned ones? Is it due to optimization landscape, overfitting, or initialization?

**Ethical Concerns:**

["NO or VERY MINOR ethics concerns only"]

**Final Justification:**

The author has addressed most of my concerns, and I will raise my score.

**Quality:**

3

**Strengths And Weaknesses:**

Strengths:

1. The paper provides regret bounds for sequential prediction methods when combined with the proposed preconditioning, advancing prior work especially for asymmetric, marginally stable linear dynamical systems. Theoretical claims are generally supported by concrete statements. The preconditioning method is mathematically principled, leveraging properties of Chebyshev and Legendre polynomials, and can be applied to a broad class of sequence models, including regression and spectral filtering.

2. The approach provides regret bounds that do not scale with the latent dimension of the dynamical system.

3. The experiments are well designed, evaluating on synthetic linear, nonlinear, and RNN-generated data. Different preconditioning schemes are compared, including learned preconditioning, and a thorough sweep across polynomial degrees and hyperparameters is conducted.

Weaknesses:

1. The assumption on the eigenvalues' arguments in Theorem 2.1 maybe too strong for complex neural networks, as it demands a system behavior with virtually no oscillations. If this assumption holds true, please cite some papers to validate this.

2. Tables 2–4 list results for several polynomial degrees, but the text could provide deeper interpretation of how performance depends on the degree parameter, especially for higher degrees where some results appear inconsistent.

3. While the proof for Algorithm 3 relies on the common theoretical assumption that the system matrix can be diagonalized (justified by density arguments), the paper could more explicitly discuss how “arbitrarily small perturbations” to achieve diagonalizability might accumulate over T steps in an online learning context.

4. The proof of Theorem C.7 misapplies Lemma C.8, which explicitly requires Krylov matrices to have Hermitian arguments. However, the matrix constructed in the proof of Theorem C.7 (diag(z)) is generally not Hermitian, as its diagonal elements are typically complex numbers. Please do not hesitate to correct me if my understanding is inaccurate.

Minor issues:

1. The table number and title always appear before the table. Table 1-4 should be revised.

2. When writing the thesis, there are few citations of related work, which affects the readability of the article. For example, Line 21-22: “It is widely acknowledged that learning this sequence is easier than learning the original sequence for a large number of modalities.”

3. Line 21: “easier", Line 127: “closed loop"

---

> ### Author Rebuttal · Authors · 2025-07-29
>
> Thank you for your insightful review! Our responses are below.
>
>
> Weakness 1)  The assumption is actually very mild since it only depends logarithmically on the horizon T. More importantly, this assumption is somewhat tight, see our discussion on this see Appendix A.3 (page 19 of our supplementary material). It’s true that as the imaginary component is restricted, the oscillatory nature of the signal is restricted. However, to our knowledge, this is the strongest regret bound possible (i.e. that isn’t polynomially dependent on hidden dimension) with the least amount of assumptions, see Appendix A.1 for a better sense of previous assumptions on linear dynamical systems in the literature. Moreover, please note that empirically we show that our method is in fact able to perform well on systems that can have oscillations.
>
>
> Weakness 2) The results are consistent with our paper but we did not make that point as clear as it could have been and we will fix that. Indeed, for Chebyshev and Legendre polynomials, once the degree is higher than 5 − 10, the performance degrades since the $\ell_1$-norm of the coefficients is very large. This is consistent with our regret bounds– see Lemma 4.2, which shows that these coefficients grow exponentially fast and see Theorem C.1 which shows that the $\ell_1$-norm of these coefficients adversely affects the regret bounds. We will make this clearer in the main body of the paper (although we do mention it in line 175).
>
>
> Weakness 3) Our results will not depend on the arbitrary small perturbations, since we have no dependence on the distance between eigenvalues, which can be very small if we start with a degenerate matrix.
> Because of density, given any fixed horizon T and any system matrix A, there is a diagonalizable matrix A’ which is no more than $1/T^{1000}$ away from A (using your favorite metric). Therefore any accumulation over T would introduce error no more than $1/T^{999}$. Note we could have picked a number different from $1000$, we just chose it for an example. Moreover, this is fixed in advance even in an online learning setting– the matrix A does not evolve with time. Does this clarify things? We are happy to add this to the paper!
>
>
> Weakness 4) This is a great catch and we thank the reviewer for finding it! Your understanding is accurate, however we do not actually need to use Lemma C.8 to show exponential decay. We just need to show our matrix satisfies a Sylvester equation like 5.2 in [1] which it does using the same matrix Q as used in the reference. The next change we must make is the argument to bound the Zolotarev number which we can easily do since the spectrum of $\mathrm{diag}(z)$ does not overlap with Q. This is true because the $z_j$ derived from Gaussian quadrature satisfies $|z_j| \leq 1 - 1/10T^2$ (this comes from the Gauss-Legendre nodes). This gap means that we will still have exponential decay in the eigenvalues, but it will be a poly log factor slower (which does not meaningfully affect our result). We will edit our proof for this but note that it proceeds exactly the same way. Please let us know if you would like to see the full edit.
> [1] Bernhard Beckermann and Alex Townsend. On the singular values of matrices with displacement 318 structure. SIAM Journal on Matrix Analysis and Applications, 38(4):1227–1248, 2017
>
>
> Question 1) This is a good question and it isn't clear. That said, the important take away is that in the previous literature you either must assume that the imaginary part is exactly 0 or that the hidden dimension isn't too large. Instead we loosen both of those assumptions to simply that imaginary part be bounded inverse logarithmically in $T$. Therefore, we broaden the class of linear dynamical systems that can be learned.
>
>
> Question 2) One major advantage to our proposed preconditioning method is that it is incredibly fast computationally-- regardless of the complexity of the algorithm being used. Indeed, it involves convolving the output sequence with just 5 scalars and is very fast in practice.
>
>
> Question 3) We randomize over initialization and optimization learning rates so it is more likely due to the fact that a) learning the best coefficients jointly with the model parameters is nonconvex and can therefore fail and b) some kind of overfitting. It's an interesting question for future work whether there are better/more robust methods to learn these coefficients.

---

> > ### Comment · Reviewer_u5xs · 2025-08-05
> >
> > The author has addressed most of my concerns, and I will raise my score.

---

### Decision · Program_Chairs · 2025-09-17

**Decision:**

Accept (spotlight)

**Comment:**

In this paper, the authors consider the preconditioning problem in sequential prediction. They propose Universal Sequence Preconditioning (USP), which applies convolution to the input sequence using coefficients of Chebyshev or Legendre polynomials.
This operation corresponds to applying a polynomial to the hidden transition matrix. For marginally stable asymmetric linear dynamical systems, the method yields the first sublinear regret bounds independent of the latent dimension (up to logarithmic factors). It also extends spectral filtering to the complex plane with theoretical guarantees.
Through numerical experiments, the authors show that the approach improves prediction across linear and nonlinear settings and learning algorithms, including RNNs.

After the rebuttal and discussion, reviewers were broadly positive about the paper. Reviewers did note serious presentation issues (for example, an equation referred to as “Eq. TO DO”), but they agreed that the paper could achieve sufficient clarity if the authors incorporate the discussion feedback in the next revision. There was also a consensus that experiments with non-synthetic data would enhance the impact of the paper. The authors appear to have already conducted some such experiments (see, for example, the discussion between Reviewer EHYC and the authors), and it is expected that these results will appear in the final version.

While the writing of the paper requires significant improvement, reviewers did not find these issues to warrant another round of review (several reviewers noted that the supplementary material is written much more clearly). At a minimum, the method provides a strong initialization for other learning methods and could have broad impact across domains. For these reasons, I lean toward acceptance.